# The Water Governance Reform Framework: Overview and Applications to Australia, Mexico, Tanzania, U.S.A and Vietnam

**R. Quentin Grafton [1],\*, Dustin Garrick [2], Ana Manero [3] 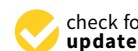 and Thang Nam Do [1]**

[1]  Crawford School of Public Policy, The Australian National University, Lennox Crossing, Canberra, ACT 2601, Australia; thang.do@anu.edu.au

[2]  School of Geography and the Environment, Oxford University Centre for the Environment, University of Oxford, South Parks Road, Oxford OX1 3QY, UK; dustin.garrick@ouce.ox.ac.uk

[3]  School of Agriculture and Environment, University of Western Australia, Perth, WA 6009, Australia; ana.maneroruiz@uwa.edu.au

\*  Correspondence: quentin.grafton@anu.edu.au; Tel.: +61410680584

**Abstract:** The world faces critical water risks in relation to water availability, yet water demand is increasing in most countries. To respond to these risks, some governments and water authorities are reforming their governance frameworks to achieve convergence between water supply and demand and ensure freshwater ecosystem services are sustained. To assist in this reform process, the Water Governance Reform Framework (WGRF) is proposed, which includes seven key strategic considerations: (1) well-defined and publicly available reform objectives; (2) transparency in decision-making and public access to available data; (3) water valuation of uses and non-uses to assess trade-offs and winners and losers; (4) compensation for the marginalized or mitigation for persons who are disadvantaged by reform; (5) reform oversight and "champions"; (6) capacity to deliver; and (7) resilient decision-making. Using these reform criteria, we assess current and possible water reforms in five countries: Murray–Darling Basin (Australia); Rufiji Basin (Tanzania); Colorado Basin (USA and Mexico); and Vietnam. We contend that the WGRF provides a valuable approach to both evaluate and to improve water governance reform and, if employed within a broader water policy cycle, will help deliver both improved water outcomes and more effective water reforms.

**Keywords:** Murray–Darling Basin; Colorado; water scarcity; IWRM; equity

## 1. Introduction

World freshwater extractions (surface and groundwater) increased by about 2.5 times from 1960 to 2010. As a result, some four billion people live in conditions of severe water scarcity where levels of water consumption are more than twice that of the readily available water, at least, one month per year [1]. Further, up to 80% of the global population is exposed to high levels of water threats in relation to watershed disturbances, pollution, water resource development and biotic factors [2].

The on-going challenge is that water extractions are projected to increase by a further 50% by 2100 [3] and, with business as usual, about half of the world's population by 2050 is projected to reside in water basins where more than 40% of the available water is extracted [4]. Using a Blue Water Sustainability Index (BIWSI), which measures the proportion of blue water use from non-sustainable water resources (including groundwater and uses that reduce environmental flows), Wada and Bierkens [3] estimate that currently some 30% of human water consumption (including groundwater) is non-sustainable in that it will result in either the degradation of surface water or depletion of groundwater resources. At a regional level, Nechifor and Winning [5] project that India

will need to lower its projected 2050 total water demand by almost 40% (292 Billion Cubic Meters, BCM), the rest of South Asia by 43% (100 BCM), the Middle East by 42% (168 BCM) and North Africa by 17% (30 BCM), if water extractions are not to exceed the current total renewable water resources of these regions.

Aeschbach-Hertig and Gleeson [6] argue that current food production in key farming regions of India, China and the USA cannot be maintained unless groundwater levels are stabilized. Another concern is that Wada et al. [7] estimate that, in 2000, non-renewable groundwater extraction contributed to 20% of global irrigation water extraction. This has important implications for food production because, globally, irrigation accounts for about 70% of global freshwater extractions and provides 40% of total human food calories. This tension between water for food production and other purposes is likely to be exacerbated into the future. For instance, by 2050: (1) the current water supply is projected to be less than the projected water applied for irrigation in major food-producing countries with production methods, and (2) a plateau is projected in terms of crop food production from water extractions if there are no further increases in the global irrigated agriculture area [8]. The key point is that in the absence of reform that includes: (1) better water governance and (2) how water is currently extracted and consumed, there are large, and with climate change, increasing risks to future food security [9].

Government responses to the global water crises have, typically, been to adopt a "hard" infrastructure or engineering solutions to increase water supply that has sometimes been part of "water nationalism" [10] and even as a "hydraulic mission" as part of a foreign policy tool [11]. Arguably, since at least the 1990s, and certainly since the Report of the World Commission on Dams was published in 2000 [12], there has been an increasing focus paid to "soft" infrastructure and governance [13]. A reprioritization towards governance and sustainability of water use is welcome as traditionally implemented water supply "solutions" have often been delivered with little regard to ecological impacts, to their negative consequences on freshwater ecosystems [14] or to the poor or marginalized who lack a voice in water planning [15]. While "hard" infrastructure is frequently needed to mitigate water insecurity, the pressures (such as population and per capita income growth, and urbanization) and states (such as per capita water availability, water variability, and climate change) of water challenges [16] require multiples responses (such as water demand management and also water justice which includes fairness, equity, participation and the democratization of water governance) [15].

Here, we provide guidance as to what should be strategic considerations in response to a growing water demand with a limited water resource and especially in terms of equity with regard to how water is allocated and used [17]. Our focus is on water governance reform noting that a change process is ongoing, must be context specific, and designed to respond to multiple challenges [18]. While there are already existing governance principles and frameworks [19–21], we contend that there is still a real need for practical guidance about how to apply key strategic considerations in relation to water reforms, and to do so in an integrative way. In Section 2, we briefly describe existing water governance frameworks and outline our own seven strategic considerations for water governance reform. In Section 3, we apply these strategic considerations in four different locations to show the added value of our approach and also discuss how our framework can be used to generate improved water outcomes. In Section 4, we offer our conclusions.

## 2. Water Governance Principles and Frameworks

Multiple frameworks and approaches exist in relation to governance, in general, and with respect to water, in particular. The Institutional Analysis and Development (IAD) framework provides a useful way of describing the broad governance space and includes: (1) the exogenous variables (biophysical constraints, community attributes and rules) and (2) the action arenas (situations and participants) that determine outcomes and feedbacks [22]. In the IAD framework, water governance reform operates within action arenas intended to influence or promote particular and desired outcomes. Within

social, economic and political settings, the IAD framework also provides a means of describing and linking resource systems, governance systems, resource units, users, interactions, outcomes and related ecosystems [23].

*2.1. Existing Water Governance Principles and Frameworks*

The most long-standing water governance framework is the Integrated Water Resource Management (IWRM) that promotes coordinated management actions in relation to environmental sustainability, economic efficiency, and social equity [24]. Beyond the principles of integration across actions and their consequences, consideration of the "triple bottom line", an exhortation to ensure participatory approaches and the full inclusion of women in management, IWRM is non-prescriptive. This has allowed IWRM to be readily adapted to multiple contexts and applied in many different ways such that 80% of countries have already adopted its principles in their water laws [25]. Nevertheless, its usefulness has been questioned [26], especially in relation to what it fails to say in regard to water resource allocation, while its dominance as a water governance paradigm is challenged by the notions of water security [27,28], the nexus [29] and integrative approaches to water policy dilemmas [30].

At an intergovernmental level, the Organization for Economic Co-operation and Development (OECD) has developed a water governance framework in collaboration with its member governments and a range of stakeholders [20]. This framework is the emerging dominant water governance paradigm given its endorsement by OECD governments. It is based on 12 principles embedded around: (1) effectiveness, in relation to defining and achieving clear and sustainable water policy goals (Principles 1. Clear roles and responsibilities, 2. Appropriate scales within basin systems, 3. Policy coherence and 4. Capacity); (2) efficiency, to maximize the benefits of sustainable water management (Principles 5. Data and information, 6. Financing, 7. Regulatory frameworks and 8. Innovation); and (3) trust and engagement, build public confidence and inclusiveness with stakeholders (Principles 9. Integrity and transparency, 10. Stakeholder engagement, 11. Trade-offs across users and 12. Monitoring and evaluation) [31]. By contrast to IWRM, the OECD framework is highly prescriptive and features: (1) a "traffic light" of the current state of water governance; (2) detailed checklists of what should be done, with a series of "What, Who and How" questions for each Principle; and a (3) ten-point assessment that involves a diagnosis and the development of an action plan to resolve the "What, When, Who and How" of implementation.

Pegram et al. [21] developed a River Basin Planning framework intended to be strategic and multidisciplinary and to deliver improved economic, ecological and management solutions at a basin scale. This framework is described using multiple examples and actual cases to show how river basin planning can be practically delivered. In common with the OECD framework, River Basin Planning is highly prescriptive and has as its core ten golden rules. These rules include: 1. Develop a comprehensive understanding of the entire system; 2. Plan and act, even without full knowledge; 3. Prioritize issues and adopt a phased and iterative approach; 4. Enable adaptation; 5. Accept basin planning is an inherently iterative and chaotic process; 6. Develop relevant and consistent thematic plans; 7. Address issues at the appropriate scale; 8. Engage stakeholders; 9. Focus on implementation; and 10. Select the planning approach and methods to suit basin needs [21]. The key steps in River Basin Planning include: (1) situation assessment, including future trends and scenarios; (2) vision formulation, including goals and outcomes; (3) Basin strategies, including conservation, water use and development, disaster risk management and institutional management; and (4) detailed implementation, including activities and milestones, responsibilities and monitoring and review [21].

An alternative to the mainstream discourse on water governance is the Framework on Hydro-Hegemony (FHH) introduced by Zeitoun and Warner [32] and developed to analyze trans-boundary water conflicts. It has been widely employed at a river basin level [33] and provides an understanding of how three forms of power ("hard" or structural power; covert or bargaining power to shape agendas; and "ideational" power to shape perceptions and discourses) are used [34] and the nature of power asymmetries. While the FHH is not a governance framework per se, it does provide

an effective means to better understand socio-political-economic relations in relation to water, and how these relations determine water outcomes.

*2.2. The Water Governance Reform Framework (WGRF)*

Given the existing water governance frameworks (IWRM, OECD, and River Basin Planning), why is there a need for a water governance reform framework? First, there is a "sweet spot" between the highly flexible, even vague, approach of IWRM and the highly prescriptive, even restrictive, rules and traffic lights frameworks of the OECD and River Basin Planning. Second, we contend the water governance reform framework (WGRF) offers both a more concise and easier-to-apply approach that complements the detailed water planning in the OECD and River Basin Planning approaches, among others, as well as more general policy frameworks. Third, as far as we are aware, the WGRF is the only water governance framework specifically developed and applied for strategic water reform. Fourth, it comprises three key strategies for integrative water security research that include: (1) linkage between the state of knowledge to decision-making; (2) an expanded water research agenda, such as comprehensive water accounting; and (3) a recognition of inequities in terms of water allocation and also the need for water justice [30].

The WGRF has as its core seven strategic considerations in relation to water reform and its implementation. Importantly, it is not a "checklist", but rather a set of strategic considerations that include: (1) well-defined and publicly available reform objectives; (2) transparency in decision-making and public access to available data; (3) water valuation of uses and non-uses to assess trade-offs and winners and losers; (4) compensation for the marginalized or mitigation for persons who are disadvantaged by reform; (5) reform oversight and "champions"; (6) capacity to deliver; and (7) resilient decision-making that is both beneficial and durable from a broad socio-economic perspective [35]. Of these strategic considerations; (3) in relation to water valuation, (5) reform oversight and "champions" and also (7) resilient decision-making are additional to the OECD and River Basin Planning frameworks. Importantly, the WGRF is also explicit about water equity in relation to (3) evaluation of winners and losers and (4) compensation for the marginalized and also those disadvantaged by reform. Thus, while the WGRF includes elements of existing frameworks, it is integrative, flexible and fit-for-purpose and, thus, a novel framework in its own right.

## 3. Applications of the WGRF to Australia, Tanzania, Mexico and USA, and Vietnam

The value of any policy framework is not in its principles or steps per se, but rather how they are applied and, importantly, whether the framework generates positive net public benefits to the alternatives. The tactical aspects of water reform must also be context-specific and, thus, the WGRF should not be applied as a step-by-step "How-To-Manual" because what is prioritized and the sequencing of water reform must differ according to values, capacity, hydrological constraints, institutions and other factors.

To show both how to apply the framework and to demonstrate its potential for decision makers, we provide four applications of the WGRF: Murray–Darling Basin (Australia); Rufiji Basin (Tanzania); Colorado Basin (Mexico and USA); and Vietnam. The choice of these applications is, in part, based on our respective knowledge and experiences. The applications were selected to ensure a large variation in terms of institutional context, history, financial resources and capacity, and biophysical differences so as to test the flexibility and applicability of the water governance reform framework. For each application, we present an overview of the biophysical and socio-economic environment, which is then followed by an evaluation of each of the seven strategic considerations.

*3.1. Murray–Darling Basin, Australia*

The Murray–Darling Basin (MDB) is located in Southeast Australia and covers an area of over one million km$^2$. The MDB suffers from highly variable rainfall and, sometimes, severe droughts. While there have been various supply-based strategies to respond to the risks of droughts, such as

the construction of large up-stream water storages, there has also been a well-recognized need to undertake water reform in terms of how water is used within the basin [36].

The most recent water reform in the MDB began with the National Water Initiative (NWI), agreed to by Basin states and the Australian government in 2004 [37]. Article 5 of the NWI highlighted the need to " . . . ensure the health of river and groundwater systems by establishing clear pathways to return all systems to environmentally sustainable levels of extraction." In addition, the NWI prioritized the establishment of consistent rules in relation to water rights and the need for comprehensive water accounting.

A lack of progress in the implementation of the NWI led to the *Water Act 2007* that reassigned the jurisdictional powers for governance of water in the MDB from the Basin states (Australian Capital Territory, New South Wales, Queensland, South Australia and Victoria) to the federal government. This act is being implemented through a ten-year Basin Plan that passed the Federal Parliament in November 2012 [38]. The 2012 Basin Plan specifies catchment and basin-level sustainable diversion limits (SDLs).

To encourage states to agree to the change water governance powers, initially opposed by the state of Victoria, and to give effect to key objects of the *Water Act 2007*, the federal government allocated A\$10 billion (subsequently increased to A\$13 billion in 2008) over ten years to compensate irrigators and ensure the success of water reform [39]. This financial reform package [36] included some A\$8.9 billion to respond to over-allocation of water by buying back water entitlements from willing irrigators (A\$3.1 billion) and also by modernizing irrigation infrastructure (A\$5.9 billion) to increase the irrigation efficiency.

(1) Well-Defined Reform Objectives

The *Water Act 2007* has well-defined, but high-level objectives. The 2012 Basin Plan gives effect to this Act and key environmental targets to be achieved beyond 2019, as detailed in the Murray-Darling Basin Authority's (MDBA) basin-wide environmental watering strategy. Key environmental targets include: (1) maintain base flow levels at 60% of natural flows; (2) enforce environmentally sustainable limits on the quantities of surface water and groundwater that may be taken from the basin water resources; (3) increase overall flow by 10% more into the Barwon–Darling, 30% more into the River Murray and 30–40% more to the Murray mouth which opens to the sea 90% of the time to an average annual depth of one meter; (4) export 2 million tons of salt per year to the Southern Ocean; and (5) improve bird breeding with up to 50% more breeding events for colonial nesting species and a 30–40% increase in nests and broods for other waterbirds [40]. While there have been some environmental improvements in specific locations [41], none of these five key objectives at a basin scale, as of the start of 2019, have been realized [36,42–46].

(2) Transparency

The Australian government's own Productivity Commission, in a five-year review of the 2012 Basin Plan, raised serious concerns about the lack of transparency and accountability in terms of institutional and water governance arrangements [47]. It observed: 'This lack of transparency has resulted in stakeholders seeking information through other means, including Freedom of Information requests and orders for the production of documents in the Australian Parliament. The absence of transparency has engendered an environment of low confidence and trust in Governments' [47]. Further, despite multiples of billions of expenditures on water infrastructure by the Australian government, there has been no publicly available cost-benefit analysis in relation to these expenditures [46].

Notwithstanding these deficiencies, water reform funds allocated for water data collection have resulted in a substantial improvement in the quality and range of data available. These data include information on water flows within the basin, such as the Water Data Online and the National Water Account, both of which are available through the Bureau of Meteorology [48].

(3) Water Valuation

A number of water valuation studies have been undertaken in the MDB, including several non-market studies [49–54]. However, a major omission in terms of quantitative valuation studies is in terms of Indigenous water values [55], notwithstanding several important qualitative studies, such as Weir [56].

Hatton MacDonald et al. [54] provided a total economic value for environmental assets located at the Murray River Mouth of A\$13 billion. Akter et al. [49] estimated marginal non-use values of water in a key Basin wetland that are comparable to the annual market price for water used for irrigation. While these non-market studies and hydrological-economic studies [57,58] have been important in making the possible trade-offs between use and non-use values of water transparent, there is no evidence they made any difference in the determination of SDLs in the 2012 Basin Plan. Indeed, sworn testimony to the Murray–Darling Basin Royal Commission (MDBRC), shows that the final determination about what should be the SDLs under the 2012 Basin Plan was dominated by political considerations with little or no regard to scientific or valuation studies [59].

(4) Compensation and Mitigation Mechanisms

Almost all the compensation paid for the reallocation of water as part of the 2012 Basin Plan has been provided to irrigators. The total compensation is A\$8.9 billion, and expenditures to date include: (1) A\$2.5 billion associated with the direct purchase of water entitlements from willing irrigators and (2) A\$3.5 billion in subsidies and grants for water irrigation infrastructure [36]. The compensation already paid represents, on average, about A\$750,000 per irrigator [60]. Given that there is no obligation for irrigators to either sell their water entitlements or to accept water infrastructure subsidies, the costs of reallocating water to the environment with the current water reform represents full compensation to irrigators for agreed to changes in their water diversions.

The approach taken to compensation and mitigation in the MDB has been to direct it to irrigators in the belief that benefits will "trickle down" to rural communities. An alternative would have been to invest up to several billions of dollars in "thriving communities" and still have had sufficient funds leftover to acquire the volumes of environmental water obtained under the actual water reform process [61]. Importantly, the Traditional Owners of the land and water of the basin, Australia's First Peoples, have received virtually no compensation for their land and water rights [62] and " . . . no material increase in water allocation for Indigenous—social, economic or cultural purposes" [63]. This is despite Article 25 of the 2004 NWI highlighting Indigenous needs in relation to water access and management, and NWI Articles 52–54 requiring water plans account for Indigenous access and native title rights to water.

(5) Reform Oversight and Champions

Reform oversight and a "champion" of change are critically important to deliver successful water reform, as shown in relation to water market reforms over the past 25 years [64]. A water reform "champion" did exist, the National Water Commission (NWC) that was created as part of the 2004 NWI, but it was abolished by an Act of Parliament in 2015. The Parliamentary Secretary responsible for announcing the NWC demise justified this decision on the basis that " . . . there is no longer a need for a stand-alone entity to undertake monitoring of Australia's progress on water reform" [65].

Despite the Parliamentary Secretary's claim that a body like the NWC was no longer needed, multiple failures in terms of implementation of the Basin Plan and delivery of key objects of the *Water Act 2007* have been identified. In particular, the Senior Counsel assisting the MDBRC, Richard Beasley, stated "The implementation of the Basin Plan has been marred by maladministration . . . The responsibility for that maladministration and mismanagement falls on both past and current executives of the MDBA and its board" [66]. This conclusion that the MDBA has not provided the reform oversight required, and that an alternative and truly independent water agency is required to audit progress

on the Basin Plan and its delivery of the *Water Act 2007*, is supported by Grafton [46], Grafton and Wheeler [36] and the Australian Government's own Productivity Commission [47], among others.

(6) Capacity to Deliver

Australia is a world leader in various water disciplines that include hydrology, hydrogeology, water law, water and climate modeling, environmental flows, and water economics, among others. This expertise resides primarily within universities, but also within government research agencies that, at a federal level, include the Commonwealth Scientific and Industrial Research Organization (CSIRO), the Bureau of Meteorology and Geoscience Australia. Scientific and technical capacity also previously existed within the NWC, and some expertise resides in state agencies and in the private sector.

In a country well-endowed with scientific and social science capacity, much of this expertise has, unfortunately, not been used in relation to key decision making with respect to the basin, and even disregarded. Key implementing agencies, such as the MDBA, have relied heavily on the reports of paid consultants for key studies in relation to water reform yet, as noted by Wheeler et al. [67], many of the socio-economic consultant reports commissioned by the MDBA suffer from serious technical deficiencies. Of equal concern is sworn testimony that a key CSIRO report used in modeling the multiple benefits of the SDLs in the 2012 Basin Plan " . . . was altered by the CSIRO management under pressure from people at the Basin Authority" and that " . . . the report was altered in a way that made it misleading" [66]. This appears to be a systemic problem, not just limited to the MDBA, as Ken Matthews, previously the head of two Australian government departments and the CEO of the NWC, has observed that " . . . current water decision-making processes have been designed with an assumption that good science and careful analysis will make its way up through the system, and that responsible ministerial decision makers will be at the helm to receive it. But it turns out that too often they are not" [68].

(7) Risk and Resilient Decision-Making

There are multiple risks that need to be managed in the context of water and the MDB. First, and foremost, is the very large variation in precipitation across seasons and years that can result in both floods and extended droughts. This risk has been managed by building very large water storages and allocating water to irrigators via water entitlements that are a share of a consumptive pool [69]. Thus, in periods of low water storages and inflows, holders of water entitlements, and in particular, those with low-reliability water entitlements, receive less than they would in "normal" years. This allocation of water also places a priority on providing water volumes to water rights with subsidiary priority on discretionary environmental flows that are not managed as water entitlements [70].

Despite a median climate change scenario developed by CSIRO for 2030 and 2050 that has, respectively, an 11% and 17% reduction in the runoff for the southern MDB [71], there was no consideration of climate change in the setting of SDLs as part of the Basin Plan [72,73]. Equally relevant, no consideration was made for the effects of climate change on the economic returns associated with the A\$3.5 billion already spent, and billions to be spent, on upgraded irrigation infrastructure [73,74]. This is problematic because modeling suggests that a more resilient strategy, in relation to droughts, is to reduce water diversions rather than invest in water infrastructure [75].

### 3.2. Rufiji Basin, Tanzania

The Rufiji River Basin is the largest river basin in East Africa, covering almost 180,000 km$^2$—or one-fifth of Tanzania's area [76]. The basin varies widely in elevation (from sea-level to 2960 m) and climate, ranging from hot and humid along the coast, to cool and moderately dry in the highlands [77]. From an administrative viewpoint, the Rufiji River Basin comprises 26 districts; while, hydrologically, the basin is divided into four river catchments: Great Ruaha, Kilombero, Luwegu and Lower Rufiji. The Great Ruaha is the largest catchment, accounting for almost half of the basin's area and over 80%

of its consumptive water uses—although it only contributes one-fifth to the 31 BCM/year of the basin's flow at the delta.

Over the last decades, management of natural resources in Tanzania has undergone a significant transformation to a decentralized system. During pre-colonial times, natural resources were managed under customary rules [78], and it was not until the early 1900s that formal water law was introduced by German and British settlers. By the 1950s, and until two decades after independence in 1961, water governance remained vested under the national government [79]. During the 1980s, widespread and poor infrastructure performance, and also negative environmental outcomes, drove a shift towards decentralization and the introduction of river basins as water governance entities. The 1990s and 2000s were dominated by a comprehensive reform, culminating in the passage of the *2002 National Water Policy* and the *2009 Water Resources Management Act* [80].

The water-governing authorities are structured on five levels of management: (i) national; (ii) basin; (iii) catchment; (iv) district; and (v) community [81]. At the national level, the Ministry of Water and Irrigation (MoWI) is Tanzania's uppermost water authority and is responsible for formulating and updating the country-wide policies. Under the MoWI, there are nine Basin Water Boards that are responsible for the allocation and protection of water resources. Beneath the Boards are Catchment Water Committees (CWC) intended to coordinate Integrated Water Resource Management (IWRM) plans and to resolve regional water conflicts. Only a few CWC have been established [80] so to overcome this water governance gap, District Facilitation Teams have emerged from administrative District Councils and are responsible for conflict resolution, water infrastructure planning, and the formation of Water User Associations (WUAs) [82,83].

(1) Well-defined Reform Objectives

Tanzania's subsidiarity principles were adopted in line with the United Nations Agenda 21, calling for water resources management to be delegated at the lowest appropriate level. Nevertheless, not all objectives defined at the national level (e.g., cost-recovery, equity and efficiency) can effectively be devolved to WUAs. In an illustrative example in the upper Great Ruaha River catchment, van Koppen et al. [84] observed how a newly introduced fee system eroded customary water-sharing principles. Thus, informal self-supply in Tanzania remains a key feature of rural water provision [85] and must be considered, alongside formal rules, to deliver an effective IWRM framework [86].

(2) Transparency

Disputes over water are common at the local level, i.e., between and within WUAs. Local conflicts can impose severe consequences for the community and beyond, as they often result in eroded cooperation among users and a lack of adherence to water rules and mismanagement of the water resources. Bureaucratic and logistic constraints also pose major obstacles to access legal water institutions by remote, ill-resourced water users [82]. Thus, transferring authority to judge water-related disputes from regional courts to local WUAs could make the legal system more accessible. Further, transparent and accountable local institutions would fill in the current gap left by the limited reach of national institutions [87].

(3) Water Valuation

Decreasing stream flows and growing water demands for environmental flows, hydropower and agriculture have fuelled tensions over water resource allocation within the Rufiji River Basin. Tanzania relies on this basin for over four-quarters of its installed hydropower capacity and it is also home to the Ruaha National Park—the country's largest—which depends on the Rufiji River as its sole water source during the dry season.

Given its importance, numerous studies on the Great Ruaha catchment have been undertaken [87–91]. Most recently, Yang and Wi [92] observed that effective water policies should be articulated by a combination of strategies that enhance environmental outcomes and are socio-economically acceptable.

(4) Compensation and Mitigation Mechanisms

Irrigation is the largest user within the Rufiji River Basin, accounting for almost 80% of its consumptive water uses—mainly for traditional, smallholder schemes. The existing irrigation area (87,000 ha) is projected to almost quadruple by 2030 [77], in line with the national Agriculture Sector Development Program targeting rural growth and poverty reduction. By contrast, independent studies conclude that viable expansions are limited [93] and, instead, a priority should be to increase the productivity of high-value crops irrigated during the dry months [94]. Proposed measures to reduce consumptive water use in the Great Ruaha River could also be accompanied by targeted social interventions to minimize or avoid negative impacts on the local communities including non-irrigation economic activities, such as livestock or off-farm work [95,96].

(5) Reform Oversight and Champions

Water reform is embedded across three contexts: (1) a dichotomy between formal and customary laws; (2) multiple government bodies nested within each other—in a succession from national to local levels; and (3) water governance across two parallel lines of authority; (i) the Ministry of Water and Irrigation that follows natural water boundaries and (ii) Local Government Authorities, defined by administrative boundaries.

Overlapping mandates across multiple water-governing institutions have contributed to rivalries over horizontally (across lines of governance) and vertically (across hierarchical levels) [77]. Thus, a designated, independent body to provide oversight and to harmonize water management across all institutions is required [82].

(6) Capacity to Deliver

Tanzania's water policies of the early 2000s represented a step-change towards IWRM goals, including water-use efficiency, irrigated crop productivity and equity of water supply. Unfortunately, as van Koppen et al. [97] argue, these reforms have failed to achieve several of their objectives, in particular, cost-recovery and alleviation of basin-level water scarcity. This is attributed to a lack of scientific analysis and poor stakeholder consultation which means that IWRM principles have, as yet, to be translated into reality.

The national versus local-level policy dichotomy is highlighted by the principle of "water equity" that is mandated by the National Water Policy 2002, Water Sector Development Strategy 2006, Water Resources Management Act 2009, National Irrigation Policy 2010 and the National Irrigation Act 2013. Despite this equity goal, many WUAs lack the knowledge, physical and technical capacity to monitor and deliver "water equity" [82].

(7) Risk and Resilient Decision-Making

Rising aquifer levels due to over-irrigation are contributing to soil-salinization across the Rufiji River Basin [98] and other agricultural areas [99,100]. Moreover, groundwater is increasingly polluted as a result of human activities, notably by nitrates from poor sanitation and fertilizer use [101–103]. Data on hydrogeology is extremely limited [104], yet it is understood that abstractions for domestic uses are growing, despite the severe health risks resulting from polluted groundwater.

One of the most critical challenges faced by the Great Ruaha catchment is the need to adapt to a variable climate, population growth and institutional changes. This means that water reforms should be framed as part of broader-reaching policies such as the education of local communities about watershed protection [87] and the cultivation of drought and salt-resistant crop varieties.

*3.3. Colorado Basin, Mexico and USA*

The Colorado River (637,137 km$^2$) is an international river shared by two federal countries: Mexico and the USA. It straddles seven states in the USA and two states in Mexico, supporting 2.23 million hectares of irrigated agriculture and 40 million people, including several Tribal Nations [105]. The

basin's multi-purpose reservoir system has the capacity to store approximately four years of annual average runoff (approximately 18.5 BCM), provides 4.2 GW of hydropower capacity, and supports a range of recreational uses, including rafting and boating. The upstream development of water resources has led to the decline of a once-vast delta ecosystem, which is now the focus of bi-national restoration efforts by the US and Mexico to secure water for base flows and pulse flows [18,106].

(1) Well-Defined Reform Objectives

Water allocation in the Colorado River Basin is governed by a complex mix of more than 100 laws, court decisions, operational guidelines, and technical rules known as the "Law of the River". The 1922 Colorado River Compact and the *1928 Boulder Canyon Project Act* established a fixed water allocation for downstream states within the US. This legal framework for interstate apportionment was confirmed in the Supreme Court decision on Arizona v. California in 1963; it requires "upper division" states (Wyoming, Colorado, Utah, and New Mexico) to deliver 92.5 BCM to the "lower division" states (Arizona, California, and Nevada) over a rolling 10-year period. It formally allocated an equivalent volume to the upper division states. Downstream delivery requirements from the upper division to lower division states are assessed on a rolling 10-year accounting period. In practice, the fixed allocation leaves the upper division states with residual flows and, hence, disproportionate exposure to hydro-climatic risks. Both divisions are responsible for Mexico's 1.85 BCM annual allocation secured under a 1944 international treaty.

The overarching reform responds to unsustainable water extractions, which are described as a "structural deficit" in which water use exceeds long-term renewable supplies. In 1999, long-term supply and demand intersected for the first time, coinciding with the beginning of an unprecedented 20-year sequence of dry years and increasing evidence from tree-rings and climate models of the potential for severe sustained drought and drying [107]. Despite over-allocation and a history of disputes, these pressures have prompted a reform period marked by institutional innovations. The states and other stakeholders in the basin have undertaken inter-related investments in institutions, infrastructure, and information to respond to the consequences of climate variability and change, starting with the development of interim guidelines for sharing surplus water among the states in 2001, and six years later, for sharing shortage.

(2) Transparency

Long-term planning and operational decision-making are guided by the Colorado River Simulation System (CRSS), a river system model that requires explicit and transparent assumptions regarding water availability, water deliveries and allocation rules [108,109]. The 2010–2012 Colorado River Basin Study used CRSS to engage stakeholders and establish a common language to navigate trade-offs associated with intensifying scarcity and shortage risks.

The reliance on CRSS for stakeholder engagement and water planning involves both strengths and weaknesses. On the one hand, the federal agency managing the reservoirs in the basin note that, as a result of the modeling system, 'transparency facilitated stakeholders being on relatively equal ground, rather than [certain parties] having an advantage' [108]. At the same time, the barriers to entry are substantial, which involves a steep learning curve that has required the capacity for building for historically marginalized groups, including environmental and indigenous stakeholders.

(3) Water Valuation

The development of water markets in parts of the Colorado River Basin has revealed the value of water in its competing uses, particularly between agriculture and urban uses. The Colorado-Big Thompson is home to the most active water market in the basin where shares of agricultural water are being leased or purchased by cities in the state of Colorado. In 2010, shares of water in the Colorado-Big Thompson project sold for approximately US\$6,900/mL (approximately US\$6.89/m$^3$, 2010 prices, [110]). More recent prices have been reported to be well over three times this amount, approaching US\$24,300/mL [111] (2018 prices). Outside of the Colorado-Big Thompson, water

markets are thin and constrained by high transaction costs and concerns about third-party effects [112]. As a consequence, the price of water established by markets or administrative decisions remains a poor reflection of the value of water in its competing uses, particularly for non-market benefits.

The non-market valuation of water in the Colorado River Basin has aimed to capture the economic value of water used for instream flows, recreational purposes and other types of ecosystem services. A 2014 report applied an ecosystem services framework to estimate the total economic value of the Colorado River, indicating economic benefits from US$56.6 to US$466.5 billion per year with an underlying asset value between US$1.5 and US$11.5 trillion [113]. The cultural values have proven difficult to integrate into decision-making, as illustrated by efforts to address lingering controversies over indigenous water justice and historical exclusion of indigenous groups from water planning and allocation [114].

(4) Compensation and Mitigation Mechanisms

Institutional mitigation and adaptations include interstate and bi-national agreements for coordinated operations of reservoir storage, together with new rules for managing surpluses and shortages, incentives for system efficiency improvements, and commitments to ecosystem restoration. Investments in infrastructure include the operation of desalination plants, conservation measures, and reservoir intakes. In the context of prolonged drought conditions, environmental flow requirements in the Delta have received additional attention, culminating in 2012 in "Minute 319" (updated in 2017 in "Minute 323"), an agreement made under the 1944 Treaty, coordinating US and Mexico's water storage and delivery options to enhance water supply reliability for Mexican water users and the Delta ecosystem [106].

Looking forward, the annual average cost of reducing shortage risks is projected to approach up to US$6 billion in 2060, reflecting a lingering bias toward supply-side solutions including the importation of water from other basins [105]. Much of these costs involves alternative water supplies (desalination) or importation, rather than improved water allocation, which is lower cost but concentrated on powerful agricultural user groups. Even restricting mechanisms to demand-side solutions will require substantial investment to sustain water security and safeguard the economic activities, urban centers, and ecosystems that depend on the river in a changing climate.

(5) Reform Oversight and Champions

The federal and bi-national structure of the basin involves multiple layers of oversight. It also creates ambiguity regarding authority and accountability [115]. The US Secretary of Interior serves as the "rivermaster" under the Colorado River Compact, and the federal government can, therefore, take unilateral action when the state governments lag. A credible threat has been issued in 2004 and 2018 as a spur to action by the state governments to negotiate rules for sharing the risks of shortage. Shortage rules issued in 2007 signaled this potential, noting that the Secretary 'shall evaluate and take additional necessary actions, as appropriate, at critical elevations in order to avoid Lower Basin shortage determinations as reservoir conditions approach critical thresholds . . . ' ([115], p. 40). On 13 December 2018, the sitting commissioner of the Bureau of Reclamation, Brenda Burman, threatened federal action on drought contingency planning if the states fail to achieve an agreement by 31 January 2019.

Paralleling the experience in the Murray–Darling and other cases, the existence of champions has proven pivotal in overcoming resistance to reform, including key organizations as well as influential policy leaders to sustain progress. The nonprofit sector, led by environmental NGOs, such as Environmental Defense and Audubon, have played a key role, facilitated by established members of policy and planning networks, such as high ranking officials in the Bureau of Reclamation and within state agencies [106].

(6) Capacity to Deliver

The USA and Mexico have significant institutional capacity, technical knowledge and financial resources to deliver on reform. In terms of institutional capacity, the basin has a nested set of governance arrangements that link local users with state, federal and bi-national arrangements. The chief constraint stems from horizontal coordination challenges between states with flashpoints of conflict between Arizona and California and between the Upper Basin states, particularly Colorado, and those downstream, particularly Arizona. Increasingly, rural-urban conflicts are posing challenges both locally and within interstate negotiations. Vested interests in the agricultural sector represent a formidable barrier to sustained progress with powerful irrigation districts in Arizona and California mobilizing to thwart changes.

Technical and human capacity are substantial, as highlighted by the Colorado River Simulation System and the development of a multi-state research consortium, the Colorado River Governance Initiative. Thus, the scientific and technical understanding of the basin has been sufficient to support the reform process. The chief impediments have stemmed from the legal uncertainties and institutional coordination issues noted above with surprisingly limited dispute about the underlying science.

(7) Risk and Resilient Decision-Making

Information on climate change, paleo-hydrology and related climate risks have played an increasing role in planning decision-making, supported by a Bureau of Reclamation funded program [109]. The severity of climate change impacts are increasingly evident, but action has lagged [107] due to the lingering distributional conflicts regarding the risks of shortages. Despite these lags, efforts to address environmental flows have expanded during sustained drought and structural imbalances highlighted by two agreements which update the international treaty between the US and Mexico to include provisions for a flood pulse to restore the Colorado River Delta. These experiences show an increasing commitment to adaptive and flexible water allocation with growing recognition of the interdependence of water users and the need to progress toward systemic resilience.

*3.4. Vietnam*

Effective water resource management plays a crucial role in Vietnam's economic development, with 80% of its GDP generated in its key river basins [116]. Despite having, on average, about 2000 mm rainfall per year and about 3500 rivers in 16 major river basins, water availability is highly seasonal and unevenly distributed across the country [117]. In turn, this contributes to severe water scarcity in some regions at particular times of the year [118].

Vietnam has over 90 million people and is rapidly industrializing and urbanizing, which is contributing to greater extractions of both surface and groundwater [116,117]. Consequently, stark trade-offs are emerging in water allocation for agriculture, industry, and households. Further, about two-thirds of Vietnam's water resources are sourced beyond its borders, such that its internal water resource availability is 4200 m$^3$ per person compared to an average of 4900 m$^3$ for South East Asia [119].

The supply of water is affected by water pollution with only 12% of domestic wastewater [120] and 25% of industrial wastewater treated before being discharged into streams and rivers [121]. As a result, untreated wastewater has polluted rivers and lakes in and around big cities and industrial zones, undermining the health and livelihoods of millions of people [116,117,119].

Water scarcity and pollution are being exacerbated by climate change. It is estimated that a sea level rise of one meter by the end of this century will displace about 11% of the population, mainly in the Red River Delta and the Mekong River Delta and along 3000 km of the coast [122]. Recent severe typhoons and storm surges, as well as other extreme weather events, such as drought and flash floods, have cost around US$1.75 billion [118] and are projected to be a major challenge in the future in the Mekong Delta that produces half of Vietnam's rice [116,118].

(1) Well-Defined Reform Objectives

Recognizing the importance of water, the Vietnam Government is reforming its water governance. Key goals are defined in the National Strategy on Water Resources in five areas: (1) water resource protection; (2) water resource exploitation and usage; (3) water resource development; (4) mitigation of water-related damages; and (5) improvement of water resource management capacity [123]. The water reform objectives are legalized by the Law on Water Resources (LWR) issued in 1998, and revised in 2012, and the Law on Environmental Protection (LEP) issued in 1993, revised in 2005 and again in 2014. The object of the LWR is to provide a legal basis for the management, protection, exploitation and use of water resources, as well as for the prevention, control and remedy of harmful effects caused by water [124]. The LEP 2014 governs environmental protection activities, including water environment protection [125].

To achieve its water reform objectives, government agencies for managing water resources have been established at both the central and provincial level. At the central level, the Ministry of Natural Resources and Environment (MONRE) was established in 2002 to separate policy development and regulation of water resources from ministries overseeing the exploitation and use of water resources for economic development, such as the Ministry of Agriculture and Rural Development (irrigation and flood control), Ministry of Industry and Trade (hydropower) and Ministry of Construction (municipal water supply and drainage) [126,127]. All of these ministries also have their branches at the provincial level.

(2) Transparency

Water resource management in Vietnam faces several transparency challenges. In particular, limited public access to data and data sharing have led to inefficient policy implementation [119]. While water information and knowledge sharing do occur, they are often blocked due to limited coordination between different levels of administration [126]. In addition, individuals and organizations with data are reluctant to publish and share data. Further, the participation of civil society, research institutions and local communities in river basin decision-making is limited [128]. While public participation in environmental impact assessment and other water decision-making processes is legally permitted [124,125], insufficient guidance and a lack of enabling policy have hindered implementation. Another key challenge of transparency is the monitoring and enforcement of environmental flows and water quality [116].

(3) Water Valuation

The principle of water allocation for high economic value use is included in the National Strategy on Water Resources [123], but in practice, water allocation and reallocation is, typically, not based on economic values; although, water pricing [129] and charges [130] have been identified as instruments for water policy. Instead, the administrative allocation of water in Vietnam is principally based on biophysical information [118]. Further, while the concept of environmental water values is part of the national law [125], few studies have been undertaken (for example, Nam and Son [131]; Do and Bennett [132]; Vo and Khai [133]) to support water decision-making [134].

(4) Compensation and Mitigation Mechanisms

Compensation for water-related damages is included in the LWR 2012 and LEP 2014, with the latter comprising two revisions to respond to concerns over compensation [125]. In particular, Article 164 regulates that the head of an organization that incurs an environmental violation is held responsible. Article 162 regulates that the statute of limitations of violation begins when the damage is detected rather than when the violation occurs. Nevertheless, compensation challenges remain including the absence of the capacity and technical knowledge to quantify losses.

(5) Reform Oversight and Champions

MONRE is mandated to oversee overall water resource management but faces challenges in fulfilling its role. This is because water policy development and implementation are fragmented across national ministries with limited coordination between MONRE, other national ministries, and provincial governments [126]. To illustrate, commissions for environmental protection were established in three major river basins (Cau, Nhue-Day and Dong Nai-Sai Gon) in 2007–2009, but they have struggled to achieve environmental targets [118]. Another challenge is cooperation between national and local levels. Dual supervisory roles of central agencies and provincial people's committees, in addition to unclear reporting mechanisms, have also limited the efficiency of water governance [119,126].

Reform oversight is challenged by inadequate transboundary cooperation mechanisms. The Mekong River Commission was formed in 1995 by an agreement between Laos, Thailand, Cambodia and Vietnam, but two upstream countries of China and Myanmar are not members. As a result, decisions on building upstream hydropower plants have not fully taken into account downstream costs in relation to livelihoods and the environment [135].

(6) Capacity to Deliver

Investment in the water sector has increased over the period 2006–2015 and Vietnam has invested more than US$6.4 million in 140 water programs with funding mainly from the state budget and international donors [119]. Yet, the annual investment requirement for water supply and sanitation alone is about US$2.7 billion, while the actual investment currently is less than 40% of this amount. Better planning and allocation, however, should help respond to the funding shortage by improving the efficiency and quality of public spending and also by attracting greater private financing [119].

A lack of staff and insufficient budget for environmental expenditure poses risks in relation to environmental water protection [134]. Inadequate water monitoring systems and limited modeling and management tools also hamper water planning and management [119]. In addition, while there is capacity in water sciences within, for example, the Vietnam Academy of Hydraulic Works, the Institute of Hydraulic Works Planning, and the Institute of Water Resources, key gaps in the social sciences remain.

(7) Risk and Resilient Decision-Making

Water risks are identified and regulated by the LEP, but a limited capacity within relevant water agencies means that some of these risks, especially in terms of water quality, are not effectively managed or mitigated. Without adequate risk assessment, the ad hoc construction of levees in the Mekong Delta began in the early 2000s and has resulted in increased flooding in the areas outside the levees, reduced soil fertility, degraded wetlands and also decreased rice productivity [136]. An adaptive response to this risk has been the 2017 Government Resolution 120, which is intended to promote flood-based livelihoods and adaptation to climate change [137].

## 4. Discussion

The four applications show that the WGRF is easy-to-apply, flexible and can be used in multiple contexts, and at both a basin and national scale. Collectively, the four cases show that, even in the absence of quantitative analysis and modeling, the strategic considerations provide a means to scope current water governance and to identify both the strengths and weaknesses of existing water reform processes.

The strategic considerations in the WGRF serve multiple purposes. First, as in the case of the Murray–Darling Basin, Australia, they provide a means to evaluate the successes and failures of reform, highlight power asymmetries and, importantly, provide constructive support for adaptive management of the reform process. Second, as applied to the Rufiji Basin, Tanzania, the WGRF scopes the existing water governance structure and, thereby, identifies future possible opportunities for water reforms. Thus, the framework can be used to devise a more effective water reform agenda, ex ante and

not just during the reform process or to rectify past mistakes. Third, as in the case in the Colorado Basin, the WGRF frames current and past actions to support dialogues in relation to transboundary institutional reforms. In particular, the framework identifies the necessity for reform oversight and how barriers to water reform might be overcome in the Colorado Basin. Fourth, for Vietnam, the framework shows it can provide an evaluation of water governance strengths and weaknesses at a national level and not just a basin scale. Further, in terms of Vietnam, the WGRF shows how framing of the water institutions, and their roles, provides the opportunities for better coordination, decision-making and improved water outcomes, especially in relation to water quality.

While the WGRF can be used as a stand-alone approach to water reform, we highly recommend that it be part of an overall water policy cycle that can be applied to problem identification and also within each policy sequence that comprises: (1) formulation; (2) adoption; (3) implementation; and (4) monitoring and evaluation. We show that the WGRF is a valuable descriptive tool of water governance and reform, but its principal contribution is to support adaptive decision making and resilient public policy. This is because successful water reform in relation to "wicked problems" [13] cannot be decided ex ante without regard to learning, new information, changes in circumstances of stakeholders or a range of other unknown or unknowable factors. It is in this context of socio-ecological systems that the WGRF facilitates evidence-informed actions and integrative decisions that support desired water outcomes. Thus, we contend that WGRF should be core to any water governance reform process and is flexible enough to be applied at both a local, basin and national scale.

## 5. Conclusions

The world stands at a critical threshold in terms of how water is extracted and consumed, by whom, when and where. Consequently, decision-makers face key challenges in terms of how to balance water supply with demand without compromising the long-term sustainability of riparian ecosystems or aquifers. This requires water conservation and water reallocation as part of an on-going governance reform process. While there are already a number of water management (such as Integrated Water Resources Management) and governance guidelines (such as OECD Water Governance Principles), we contend that none provide the "sweet spot" in terms of ease of use, flexibility to multiple scales and contexts, or is integrative in relation to the reform research agenda, especially in relation to inequities in water allocation.

In response to the needs of decision-makers in relation to water allocation and water outcomes, we developed the water governance reform framework (WGRF). It is a strategic framework that allows both stakeholders and decision makers to review seven key considerations: (1) well-defined and publicly available reform objectives; (2) transparency in decision-making and public access to available data; (3) water valuation of uses and non-uses to assess trade-offs and winners and losers; (4) compensation for the marginalized or mitigation for persons who are disadvantaged by reform; (5) reform oversight and "champions"; (6) capacity to deliver; and (7) resilient decision-making. In four very different applications spanning five countries, we show how the WGRF can be readily applied to provide valuable insights about water governance and the water reform process, even in the absence of quantitative analysis or modeling. We contend that these applications show that the WGRF is fit for purpose and adds important integrative features to existing governance principles. In our view, if the WGRF is employed within a broader water policy cycle, it will help deliver both improved water outcomes and more effective water reforms.

Glossary (from Grafton [13]):

**Water Scarcity**: A measure of water use to water availability. A commonly used measure is the ratio of the annual water extracted in a given location to the annual renewable fresh water available.

**Water Security**: According to the United Nations it is 'The capacity of a population to safeguard sustainable access to adequate quantities of acceptable quality water for sustaining livelihoods, human well-being, and socio-economic development, for ensuring protection against water-borne pollution, and water-related disasters, and for preserving ecosystems in a climate of peace and political stability.'

**Water Stress**: A measure of per capita annual renewable fresh water availability. Water availability of less than 1000 m$^3$ is considered to be high water stress and less than 500 m$^3$ is defined as extreme water stress.

**Author Contributions:** Authorship is alphabetical after the first author. Conceptualization (R.Q.G.); Applications (R.Q.G., D.G., A.M. and T.N.D.); Writing-Review & Editing (R.Q.G., D.G., A.M. and T.N.D.).

**Funding:** D.G. received partial funding from Social Sciences and Humanities Council of Canada. Grant Number: 430–2014-00785. A.M. received partial supported by the Australian Centre for International Agricultural Research under project number FSC2013-006.

**Acknowledgments:** The authors are most grateful to Arjen Hoekstra for the invitation to submit an article to the special 10th anniversary issue of Water.

**Conflicts of Interest:** The authors declare no conflict of interest. The funders had no role in the design of the study; in the collection, analyses, or interpretation of data; in the writing of the manuscript, and in the decision to publish the results.

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
