# Peer review of "The Water Governance Reform Framework: Overview and Applications to Australia, Mexico, Tanzania, U.S.A and Vietnam"

_water, doi:10.3390/w11010137_

Round 1

Reviewer 1 Report

This paper makes a useful contribution by setting out a framework for comparative analysis of water reforms. The proposed framework complements other conceptual frameworks being more open ended than some and less prescriptive than others. While most evaluation of water reforms are without doubt qualitative and descriptive having a common taxonomy and conceptual framework makes the comparative work simpler and more accessible. The framework provides some useful guidance for policy makers , evaluators and historian of water reforms, pointing to some of the key features of good governance of reform processes. The four worked case studies of the framework's application are illustrative and interesting in their own right. The main values and application of of the proposed approach are clearly defined. The paper is well written and deserves to be published as with, after minor copy editing and proof reading. It is a pleasure to read such a well prepared manuscript, that was clear, accessible and made sense of am important subject.

Author Response

We are grateful for your insightful comments and we are especially pleased that you find that Water Governance Framework and the four applications useful. We have copy edited the manuscript and also made additional revisions in response to Reviewer 2.

Reviewer 2 Report

The manuscript suggests a framework for the water governance reform, and applies it to selected case studies. The manuscript is overall well structured, but its flow and argument would benefit from and require revision in order to make it able to meet the standard of the articles published in Water MDPI Journal. Here are comments that the authors should consider when revising the manuscript:

- 31-33: "World freshwater extractions increased by about 2.5 times from 1960 to 2010 such that some 4 billion people live in conditions of severe water scarcity at least one month per year [1] and up to 80% of the global population is exposed to high levels of threat to their water security [2]." Not sure if it is an English problem here or if the authors are not being clear, but I do not understand what they mean (and this is the first sentence!): so they argue that the water extraction increased causing more water scarcity etc.? I do not see a direct link and actually one would expect the opposite... please clarify. 

- 34-36: drop the reference and mention to the WEF: it is not useful to look at the global level not considering nuances around the globe, it seems that Sweden is having the same problems faced by Yemen on water scarcity. 

- 31-40: Clarify that with water extractions you mean groundwater (or not?). 

- 38-41: the link of food production and water sustainability is not nuanced and does not consider what kind of sustainability (define the term also); economically sustainable? Sustainable for whom? Are we only looking at water in its own box, without considering the interlinkages with water and food security? It is too easy to use concepts of virtual water to call for reforms in the food sector, we need however to introduce also concepts of food sovereignty and unpack the complex dynamics around water and food security. As it stands now, this is very shallow. 

- 45-46: groundwater accounted for the water demand? 

- 48-50: I'd suggest reading the latest article of Conker (2019) in Sustainability on "Peace at home, peace abroad?" on the hydraulic mission of the state of Turkey, which seems relevant to the "hard" solutions argument. Also, the work of Menga, and of the special issue 2 volume 11 of Water Alternatives. 

- 58: good to see the focus on water injustice too. However, one would expect a definition, as well as reference to the latest work of the London Water Research Group.

- Section 2: why focusing only on IWRM and OECD? That's quite reductionist, given the range of frameworks available out there. I'd suggest a better review of all frameworks of water governance. 

- Section 2.2.: this is a key section; however, i'd require this section to be expanded in order to better explain and justify why this new framework is needed, and how it differs from previous ones. From what I read in this section, it seems that it could be seen within the previous ones, and this would not really justify a new theoretical contribution to knowledge. Hence, the authors should pay more effort to this crucial section. 

- 137: so maybe we can see it not as a new framework, but as an application of existing ones? 

- 144-146: the choice of these cases needs to be justified right here, when first mentioned. Also, as emphasised by Hussein, H. & Grandi, M. Int Environ Agreements (2017) 17: 795. https://doi.org/10.1007/s10784-017-9364-y  , the authors should mention the necessity of considering the broader context (as Ostrom would say, the exogenous variables) when discussing water governance. 

- Water scarcity is never defined throughout the text; which indicator are you using, and why? Also please show the limits of the indicators (including the one you will be adopting). 

- 629 and throughout the text, there is a tendency of emphasising "water risk", "water scarcity" (as material rather than socially constructed.. look at the writing of Mehta, Allouche, and Hussein on this), as well to a tendency to consider water conflict and go into the neo-malthusian relationship of pop. growth and water scarcity (see your intro); please re-read this important contribution: https://entitleblog.org/2018/03/22/why-are-water-wars-back-on-the-agenda-and-why-we-think-its-a-bad-idea/ 

- Overall, I would also suggest including perspectives of critical water governance into this manuscript, which would strengthen the argument and also widen the audience, interest, and visibility of the contribution. Inclusion of power asymmetries consideration, as developed by Warner/Zeitoun/Mirumachi (see hydro-hegemony and TWINS), of Allan (see virtual water), of Mehta/Allouche/Hussein (see the construction of the discourse of water scarcity in India and in Jordan), would be beneficial and would allow talking to a wider audience. 

- Discussion and conclusion are very brief and should be expanded. 

Author Response

Please see attached responses.

Round 2

Reviewer 2 Report

The article seems much improved from the previous version, well done! 

Only two remaining comments:

1) the article of Conker i mentioned is the following one (I read the pre-print in researchgate if I'm not wrong), but it is now available on here: https://www.mdpi.com/2071-1050/11/1/228

2) yes, a glossary of your definitions would be useful (even if they are not the same ones of the authors you cite)

Author Response

Thank for the very rapid turn round and additional suggestions. In response we have:

Included the citation to Conker and Hussein (see line 63(

Added a short glossary based on the definitions in Grafton (2017).

Water EISSN 2073-4441 Published by MDPI AG, Basel, Switzerland RSS E-Mail Table of Contents Alert
Back to Top